# Updating and Refining of Economic Evaluation of Rotavirus Vaccination in Spain: A Cost–Utility and Budget Impact Analysis

**DOI:** 10.3390/v16081194

**Published:** 2024-07-25

**Authors:** Iñaki Imaz-Iglesia, Montserrat Carmona, Esther E. García-Carpintero, Lucía Pedrosa-Pérez, Alejandro Martínez-Portillo, Enrique Alcalde-Cabero, Renata Linertová, Lidia García-Pérez

**Affiliations:** 1National Health School, Carlos III Institute of Health (Instituto de Salud Carlos III), 28029 Madrid, Spain; imaz@isciii.es (I.I.-I.);; 2Chronicity, Primary Healthcare and Health Promotion Research Network (Red de Investigación en Cronicidad, Atención Primaria y Promoción de la Salud/RICAPPS), 28029 Madrid, Spain; eegarcia@isciii.es (E.E.G.-C.); renata.linertova@sescs.es (R.L.); lidia.garciaperez@sescs.es (L.G.-P.); 3Health Technology Assessment Agency, Carlos III Institute of Health (Instituto de Salud Carlos III), 28029 Madrid, Spain; lpedrosa@isciii.es; 4National Centre of Epidemiology, Carlos III Institute of Health (Instituto de Salud Carlos III), 28029 Madrid, Spain; ealcalde@isciii.es; 5Center for Biomedical Research in Neurodegenerative Diseases (CIBERNED), 28029 Madrid, Spain; 6Assessment Service of the Canary Islands Health Service (Servicio de Evaluación del Servicio Canario de Salud/SESCS), 38109 Tenerife, Spain

**Keywords:** economic evaluation, rotavirus vaccine, cost–utility, cost-effectiveness, budget impact, Markov model

## Abstract

Two vaccines against rotavirus diseases, Rotarix^®^ and RotaTeq^®^, are being marketed in Spain; but rotavirus is not presently among the diseases covered by universal vaccination in Spain. The aim of this study was to assess the efficiency of extending Spain’s current targeted rotavirus vaccination strategy including only preterm babies, to a policy of universal vaccination. A de novo cohort-based Markov model was built to evaluate the efficiency of three compared rotavirus vaccination strategies in Spain: targeted, universal, and no vaccination. Using Rotarix^®^ or RotaTeq^®^, we compared the cost–utility of these strategies from both a societal perspective and Spanish National Health System (SNHS) perspective. The model represents the most important clinical events conceivably linked to rotavirus infection. Efficacy, effectiveness, safety, costs, and utilities were identified by systematic reviews. Incremental cost–utility ratio (ICUR) is EUR 23,638/QALY (Quality-Adjusted Life Year) for targeted vaccination with Rotarix^®^ compared with no vaccination. The ICUR for the rest of the strategies evaluated are above EUR 30,000/QALY. The sensitivity analysis shows price as the only parameter that could make the universal vaccination strategy efficient. Considering a threshold of EUR 25,000/QALY, only targeted vaccination with Rotarix^®^ would be efficient from societal perspective. Price drops of 36.9% for Rotarix^®^ and 44.6% for RotaTeq^®^ would make universal vaccination efficient.

## 1. Introduction

Rotavirus is the leading cause of severe, dehydrating diarrhea in children aged less than 5 years globally [1]. Although the infection is a global problem, the severity and incidence are higher in developing countries, but Spain and the European countries still have an important incidence and hospitalized cases occur [2].

Since 2006, two rotavirus vaccines, Rotarix^®^ and RotaTeq^®^, have been marketed in Spain; but rotavirus is not among diseases covered by universal vaccination. According to data issued by the International Vaccine Access Center (IVAC) in April 2022, universal rotavirus vaccination has been introduced into immunization programs in 114 countries, with 18 countries planning to introduce such programs, and 62 countries having neither introduced nor indicated plans to introduce them within the next 3 years [3]. In November 2019, Spain decided to finance this, but solely for premature babies born between 25 and 32 weeks of gestation, due to their increased risk of serious complications. In Europe, only The Netherlands and Spain have adopted this rotavirus vaccination strategy [4]. The Spanish Agency for the Evaluation of Medicines and Medical Products (Agencia de Evaluación de Medicamentos y Productos Sanitarios) is responsible for authorizing medicines for marketing in Spain. The General Directorate of the Common Portfolio of Services of the National Health System and Pharmacy is in charge of deciding whether medicines and medical products are included in the public funding of medicines and medical products.

In 2014, we published a cost–utility evaluation of universal rotavirus vaccination in Spain [5], which indicated that a universal vaccination with RotaTeq^®^ would not be efficient in Spain at that time with the included parameters. This evaluation has become outdated, because it did not include a targeted vaccination strategy as comparator, but also because it considered only RotaTeq^®^, but not Rotarix^®^. At the time of the 2014 publication Rotarix^®^ was suspended in Spain because of safety concerns that were resolved, and commercialized again, from July 2016 [6].

We have revisited the 2014 evaluation, updating the epidemiological information, including the latest evidence on vaccine safety, efficacy, and effectiveness, and refining the model to include nosocomial infections and risk of adverse events, extending the follow-up to lifelong follow-up, and covering the two vaccines available in Spain. This research provides with updated information for health authorities in Spain to consider different strategies regarding rotavirus immunization in Spain. The aim of this study was to evaluate the efficiency of the ongoing targeted vaccination strategy in Spain compared with no vaccination and with universal vaccination, from the healthcare and societal perspectives. In addition, a secondary objective is to conduct a budgetary impact analysis of a scenario of including universal vaccination against rotavirus in the vaccination schedule financed by the public health system with the results obtained from the cost–utility analysis.

## 2. Materials and Methods

We built a de novo cohort-based Markov model to assess the cost–utility of the following three rotavirus vaccination strategies in Spain:No vaccination;Universal vaccination;Targeted vaccination, solely for preterm babies born between 25 and 32 weeks of gestation.

We compared the cost–utility of these strategies, using Rotarix^®^ or RotaTeq^®^, approached from both a societal and a Spanish National Health System (SNHS) perspective. The model represents the most important clinical events conceivably linked to rotavirus infection. Figure 1 shows the decision tree of the no-vaccination strategy. The other two strategies use the same decision tree but also include a second branch for subjects who, for different reasons, were finally not vaccinated under the strategy. The targeted vaccination strategy is divided into two branches: vaccinated and non-vaccinated babies. Here, the “vaccinated” are preterm babies, comprising a high-risk population, and the “non-vaccinated” are non-premature babies, comprising a low-risk population.

This is a Markov model of annual cycles that follows a hypothetical cohort of all newborns in Spain until the end of their life course, set at an age of 100 years for study purposes. In the model, all newborns start in the “no previous infection” and “no vaccination” states and can evolve annually through the tree, remaining in the same state or moving to the other two states, i.e., “post-infection” or “death”.

### 2.1. Incidences and Probabilities

As rotavirus is not an infection that is subject to compulsory surveillance, the most accurate information on rotavirus incidence in Spain comes from the *Conjunto Mínimo Básico de Datos/CMBD* (Minimum Basic Data Set) [7]. This database does not furnish information on second or third infections; as a consequence, the model incorporates all the epidemiological data as rotavirus cases but not as infected persons. The CMBD is an exhaustive registry of hospitalizations in Spain. It can be used to estimate the annual incidence of Rotavirus Acute Gastro-Enteritis (RV-AGE), which accounts for all RV-AGE cases, whether the main cause or the non-primary cause of hospitalization.

In addition, other data sources were used to identify Spanish incidence of rotavirus infections and to calculate probabilities (Appendix A). The incidences were transformed into probabilities, using the following formula:1 − e ^−rate × time^

Although most of the data required to populate the model were identified, some uncertainties remained. Expert advice was taken on the assumptions, whose impact on the model’s results were then evaluated in the sensitivity analysis.

Assumptions:A 50% underdiagnosis of RV-AGE hospitalization. While the precise percentage of underdiagnosis is unknown, some must be assumed, because the diarrhea causative agent is not identified in many hospitalized cases [8,9,10].The application of the Parashar model [11] to the Spanish setting [12] shows that rotavirus cases requiring a medical visit account for 20% of all cases. The remaining 80% do not require a healthcare visit, being cared for at home.Hospitalized cases have previously received primary or emergency care.Mortality in the non-infected population is that of the general population. Mortality due to rotavirus is so low that a possible overestimation of mortality in the non-infected population can be considered negligible.A vaccine coverage of 94.7% was used in the base case, with changes in the sensitivity analysis. This was the observed vaccine coverage in Spain in 2019 for the following universal vaccinations: poliomyelitis; diphtheria; tetanus; pertussis; hepatitis B; and meningococcus B [13].The vaccine coverage of the at-risk population was expected be higher. We used Bruijning et al.’s estimate [14], which indicates a 2.3% higher vaccination coverage.The proportion of preterm babies among all newborns was estimated using data sourced from the Spanish National Statistics Institute (*Instituto Nacional de Estadística/INE*), which estimates that, from 2017 to 2019, 0.9% of deliveries occurred before week 32 [15].Rotavirus hospitalizations and emergency visits were higher in the at-risk population (OR: 2.8) [16].Rotavirus nosocomial infections were more likely in the at-risk population (OR: 2.6) [17].

The efficacy, effectiveness, and safety of both vaccines were obtained through systematic reviews (Appendix A). While the systematic review on the safety of both vaccines indicated that an increase in vaccine-related adverse events had not been demonstrated, some observational studies nevertheless reported varying information (Appendix A). In light of this, we assessed the potential influence of the risk of intussusception in the sensitivity analysis. The highest value of the range considered was that observed in the technical file of the European Medicines Agency (EMA) (a higher risk of intussusception due to vaccination of 6 cases per 100,000 vaccinated babies) [18].

### 2.2. Costs

Systematic reviews described in Appendix A were helpful in identifying cost information. In addition, searches were extended to other Spanish cost-specific data sources, as described in Table 1. Indeed, all data on costs were drawn from Spanish sources. We identified RV-AGE-related healthcare costs, as well as non-healthcare and indirect costs.

The healthcare costs included in the analysis from a societal perspective were those for oral serum and anti-diarrheal drugs for mild cases. Non-healthcare and indirect costs included were transportation, extra diapers, caregivers at home, and parental productivity losses (Table 1) [7,19,20].

**Table 1 viruses-16-01194-t001:** Healthcare, non-healthcare, and indirect costs associated with rotavirus infection (figures are in EUR and updated in September 2021).

		Direct Costs			Indirect Costs	TOTAL
		Healthcare Costs. SNHS Perspective	Healthcare Costs. Societal Perspective	Non-Healthcare Costs. Societal Perspective	Societal Perspective	
Primary care	Primary value	17 ^a^	10 ^a^	14 ^a^	125 ^a^	166
	Year	2004	2004	2004	2004	-
	Updated value	22.56	13.27	18.58	165.88	220.29
Emergency care	Primary value	204 ^a^	16 ^a^	17 ^a^	172 ^a^	409
	Year	2004	2004	2004	2004	-
	Updated value	270.71	21.23	22.56	228.24	542.74
Hospitalization	Primary value	2163.61 ^b^	14 ^a^	10 ^a^	279 ^a^	2466.61
	Year	2013	2004	2004	2004	-
	Updated value	2315.06	18.58	13.27	370.23	2717.14
Nosocomial	Primary value	744 ^c^	14 ^a^	10 ^a^	152.8 ^c^	920.8
	Year	2006	2004	2004	1999	-
	Updated value	924.79	18.58	13.27	238.98	1195.62
Home-based care	Primary value	-	5 ^a^	7 ^a^	23.8 ^c^	35.8
	Year	-	2004	2004	1999	-
	Updated value	-	6.635	9.29	37.22	53.145

^a^ Giaquinto 2007 [20]; ^b^ CMBD 2010–2015 [7]; ^c^ Díez-Domingo 2010 [19].

Other costs incurred were vaccine acquisition and vaccine administration costs. Shown below are the current sales prices in Spain, which are the same for both private and public (governmental) procurement.

Rotarix^®^ full vaccination, requiring two doses, costing EUR 187.32.RotaTeq^®^ full vaccination, requiring three doses, costing EUR 208.50.

Administration costs were taken from the analysis performed by the Spanish Ministry of Health, which estimated an average cost of EUR 6 to administer a vaccination dose [21]. As the two Rotarix^®^ doses (months 2 and 4) coincide with other vaccines included in the Spanish routine vaccination schedule, we reduced the cost by half. In the case of RotaTeq^®^, the first two doses would be in the same months, thus making the cost incurred EUR 3, but the third dose would require a specific visit and so the cost incurred would be the full EUR 6.

### 2.3. Utilities

The systematic reviews provided us with enough information on utility losses on each RV-AGE case included in the model. We incorporated utility losses through Quality-Adjusted Life Years (QALY), including infected children and two caregivers. This is a common methodologic approach in rotavirus infections because most cases are in children and QALY losses often affects both parents.

We used the estimates of Marlow et al. [22] for QALY losses among children and their caregivers for high and moderate severities. QALY losses among children with mild cases have been estimated by Aidelsburger et al. [23], who provide QALYs for children under and over 18 months old: the median between these two values was used. Since caregiver QALY losses related with mild cases were estimated by Hansen-Edwards et al. [24], without distinguishing between primary and secondary caregiver QALY losses, we used the same value in both cases (Table 2).

### 2.4. Analysis

We performed a cost–utility analysis of a potential cohort of newborns of 400,000, which is a proxy of the average of newborns in recent years in Spain [25]. The Incremental Cost–utility Ratio (ICUR) was calculated to estimate the average cost per person needed to obtain an additional QALY for a given vaccination strategy as against others.

We have adopted the ICUR threshold used in the reports of the Spanish Network of Health Technologies Assessment Agencies (*Red Española de Agencias de Evaluación de Tecnologías Sanitarias y Prestaciones del Sistema Nacional de Salud/RedETS*), which is below EUR 25,000/QALY [26]. The discount rate was 3% for both costs and utility, as recommended by the Spanish Economic Health Technology Assessment Guide [26]. All costs were updated to September 2021 and calculations included a half-cycle correction. For analysis purposes, we used the TreeAgePro 2018^®^ software program.

In addition, we also performed a univariate deterministic sensitivity analysis for all the variables of the model (52 parameters). Ranges used in the sensitivity analysis were mainly those found in the same literature as that which was selected for the base case (see Appendix A). When data on ranges were not available, a reduction of 60% and an increase of 100% were used. Appendix A shows the results of the sensitivity analysis for the comparison between the targeted and the universal vaccinations, but all variables were tested and those with the most relevant variations in the ICUR were selected to be shown in the Tornado diagrams. The results of the sensitivity analysis on including non-vaccination as a comparator are available upon request.

Lastly, we also calculated the budget impact of each of three vaccination strategies, considering the model’s results for the first five years of life, since the costs of an annual cohort of newborns over five years are similar to the expenses of the whole population in a year. This is in line with the five-year calculation period recommended by budget impact analysis guidelines [27,28]. The average annual budget that the SNHS would need to implement for each of the strategies was likewise calculated.

## 3. Results

### 3.1. Base Case

The results of the application of the model to the cohort with the two vaccines are shown in Table 3 and Table 4, respectively. These tables show the number of infections and related clinical events produced over a lifetime with each of the strategies and each vaccine.

The analysis showed that, while targeted vaccination would prevent very few clinical events, universal vaccination would prevent significant numbers of clinical events. These findings were similar for both vaccines.

Figure 2 shows the results of the cost–utility analysis of both vaccines from a societal perspective. Compared with no vaccination, targeted vaccination led to minimal changes in terms of costs and utility with both vaccines. Universal vaccination was costlier but yielded little benefit in terms of QALYs.

Costs and utility are shown in Table 5 from a societal perspective and in Table 6 from an SNHS perspective. In both tables, costs are much higher with universal vaccination than with targeted vaccination, and slightly higher with RotaTeq^®^ than with Rotarix^®^. Average QALYs were slightly higher with universal vaccination than with targeted vaccination, and almost the same with both vaccines.

Considering the proposed threshold for Spain, the base-case analysis indicated that only targeted vaccination with Rotarix^®^ compared with no vaccination is efficient from a societal perspective (ICUR = EUR 23,638/QALY). Neither of the other strategies was efficient from either perspective in Spain.

### 3.2. Sensitivity Analysis

The results of the sensitivity analysis can be seen in Appendix A. Ten variables gave rise to higher variations in the ICUR and are shown in Tornado diagrams. That said, however, the only variations that yielded a result below the threshold were price variations. The range in the efficacy/effectiveness values was wide but did not achieve an ICUR below the threshold. Another relevant variable in the sensitivity analysis was “p_low_risk_home_care”, which is the probability of having a rotavirus infection treated at home in the low-risk population under the targeted strategy. Although wide variations in this variable failed to achieve an ICUR below the threshold for Spain, the remaining variables proved increasingly less able to produce changes in the ICUR (Figure 3).

The prices that led to an affordable ICUR for extending the current strategy to one of universal vaccination were as follows: Rotarix^®^ EUR 118.20 complete vaccination (price drop of 36.9%); RotaTeq^®^ EUR 115.50 complete vaccination (price drop of 44.6%). Targeted vaccination with RotaTeq^®^ would be efficient at a price of EUR 192.00 for the full schedule, amounting to a drop of 7.9%.

### 3.3. Budget Impact

The budget impact of extending targeted to universal vaccination is shown in Table 7. Based on current vaccine prices, the net annual budget that the SNHS would have to allocate to extending the current strategy to one of universal vaccination would be EUR 49.3 million with Rotarix^®^ and EUR 56.7 million with RotaTeq^®^. On the other hand, if the prices were reduced to allow for an affordable ICUR, the annual budget for universal vaccination in Spain would fall to EUR 28.7 million with Rotarix^®^ and EUR 29.1 million with RotaTeq^®^.

## 4. Discussion

The refinement and updating of the previous economic evaluation have yielded different results. In the first place, all the comparisons have changed, owing to the inclusion of targeted vaccination and the Rotarix^®^ vaccine along with RotaTeq^®^. Targeted vaccination, which is the current vaccination strategy in Spain, includes preterm babies and those diagnosed as high-risk by a pediatrician.

Our previous evaluation did not find efficient the only comparison made, which was universal vaccination with RotaTeq^®^. Now with the current model and parameters, a targeted vaccination strategy using Rotarix^®^ was the only approach that proved to be efficient and with RotaTeq^®^ the ICUR was slightly over the threshold. It should be explained here that, as there is no official willingness-to-pay threshold in Spain, we used that recommended by RedETS [26].

Even though targeted vaccination was the only efficient strategy with current parameters, the impact of this strategy in terms of clinical events avoided is much lower than that of universal vaccination. Hence, the sensitivity analysis could be useful in identifying parameter changes that could make universal vaccination efficient in Spain. The sensitivity analysis showed consistency with most of the variables: only efficacy/effectiveness, home-care cases, and vaccine price achieved relevant changes in the ICUR. The efficacy/effectiveness values (Appendix A) displayed a wide variation, with ranges of 0.46 to 0.87 for protection against RV-AGE. This wide range was tested in the sensitivity analysis, but universal vaccination was not efficient, even with the higher levels.

Another relevant variable was the probability of a home-care case, particularly in targeted vaccination. This is a variable with a high impact in the model, due to the number of children who develop a mild case of rotavirus which only requires home care. Nevertheless, changes in this parameter, even with the wide range considered, failed to produce a relevant change in the ICUR.

Exploring the sensitivity analysis to find modifiable parameters that could make universal vaccination efficient in Spain led us to the conclusion that reducing the prices by a relevant proportion could make universal vaccination efficient in Spain. The previous assessment quoted an estimated price for RotaTeq^®^ of EUR 63, in order for universal vaccination to be efficient as against no vaccination. However, a more useful comparison in Spain is targeted vs. universal vaccination, and this result was not given in the previous assessment [5].

The current model provides a more accurate, in-depth analysis, updates the evidence, incidences, probabilities, costs and utility, and relies on Spanish data for most of the parameters. For the most part, other assessments performed in developed countries have arrived at a positive conclusion about the efficiency of universal rotavirus vaccination. In the Appendix A, a summary of the 16 identified evaluations performed in developed countries is available. Only 2 out of 16 evaluations did not find universal vaccination efficient. Moreover, most European countries have included universal vaccination in their public health benefits.

Only one study with a similar research question was identified. This is a Dutch evaluation that also compared targeted, universal, and no-vaccination strategies [14], but their targeted strategy is wider than that covered by the Spanish strategy: within the targeted population, they include all newborns before 36 weeks, weighing less than 2500 g or with complex chronic conditions. Whereas they estimate the size of the targeted population at around 7.9%, our estimate put it at around 0.9%. The Dutch assessment concludes that targeted vaccination is more efficient than universal or non-vaccination strategies. It also identifies price as the key modifiable variable but does not provide a price threshold. However, it is possible that targeted vaccination may involve higher costs because the Autonomous Regions in Spain would obtain higher negotiated prices than with universal vaccination because the purchase of vaccines would be of fewer units.

In terms of efficiency, our assessment showed Rotarix^®^ as having a slight advantage over RotaTeq^®^, even though the efficacy/effectiveness results for cases requiring assistance and severe cases were better for RotaTeq^®^. The explanation for this presumably lies in higher costs incurred in administration and vaccine prices.

Another relevant result of our assessment was the difference observed between the results for a societal versus SNHS perspective. Our estimate based on a societal perspective includes non-healthcare and indirect costs, which is more complete. The importance of indirect costs in rotavirus infections has been routinely highlighted [2,10,29,30]. This is mainly related with the importance of caregivers in childhood infections and was taken into account in our model in the estimation of both costs and QALY losses of primary and secondary caregivers, as is recommended in the literature on the rotavirus burden of disease [20,22,24]. As well as having proven its effectiveness in the real world, recent studies have shown that an optimal launch of rotavirus vaccination can generate substantial economic gains over time [31,32,33]. Even in Spain, studies have shown that hospitalizations caused by rotavirus are reduced as rotavirus vaccination coverage increases, which can also translate into cost savings for the NHS [34].

As regards the budget impact, the healthcare system would have to invest a substantial sum of money, but the benefits in terms of clinical events avoided would be high. The price reductions obtained in the sensitivity analysis would make it possible to reduce the budget impact by approximately half. The extension of the current targeted vaccination strategy to a universal vaccination strategy would not require major organizational changes, since Rotarix^®^ could be administered together with other vaccines under the current Spanish vaccination schedule; in the case of RotaTeq^®^, this would only require an additional visit at the time of the third dose.

This evaluation study has some limitations. First, the estimation of parameters is always a challenge. Most of the data on the 52 parameters included in this paper were essentially sourced from registries and databases, thereby entailing the inherent difficulties of information systems. One of these parameters is the incidence of acute RV gastroenteritis; in our study, this is assumed to be infections leading to hospital admission, but it may be underestimated acute cases without hospital admission; though it is important to note that the proportions resulting from Parashar’s model, which aims to include all cases that are not registered in the health system, have been used as a reference [11]. Second, our model does not estimate herd immunity, being a Markov cohort model rather than a dynamic model. Nonetheless, Spanish incidences could be estimated with a high degree of accuracy, because Spanish data systems, such as the CMBD, are extremely comprehensive [7].

## 5. Conclusions

It is important to highlight the fact that this is the first nationwide study to estimate the cost–utility of targeted and universal rotavirus vaccination, from both a societal and SNHS perspective, using the two vaccines currently marketed in Spain.

At current vaccine prices, only targeted vaccination of the at-risk population with Rotarix^®^ would be efficient in Spain.Compared to a non-vaccination scenario, universal vaccination would achieve a much greater impact than targeted vaccination would in terms of reducing clinical events.The sensitivity analysis performed with all the model parameters indicates that the economic modeling was robust, as there were no variables, except price, which, within their ranges, modified the results substantially.Universal vaccination would be efficient with Rotarix^®^, with price drops of 36.9 to 40.3% for full vaccination, and with RotaTeq^®^, with price drops of 44.6 to 47.9% for full vaccination.At current prices, the implementation of universal vaccination would have an annual budgetary impact for the NHS of between EUR 49.3 and EUR 56.7 million, depending on whether vaccination was performed with Rotarix^®^ or Rotateq^®^. With the price drop resulting from the sensitivity analysis, the budget impact would be less than half.

## Figures and Tables

**Figure 1 viruses-16-01194-f001:**
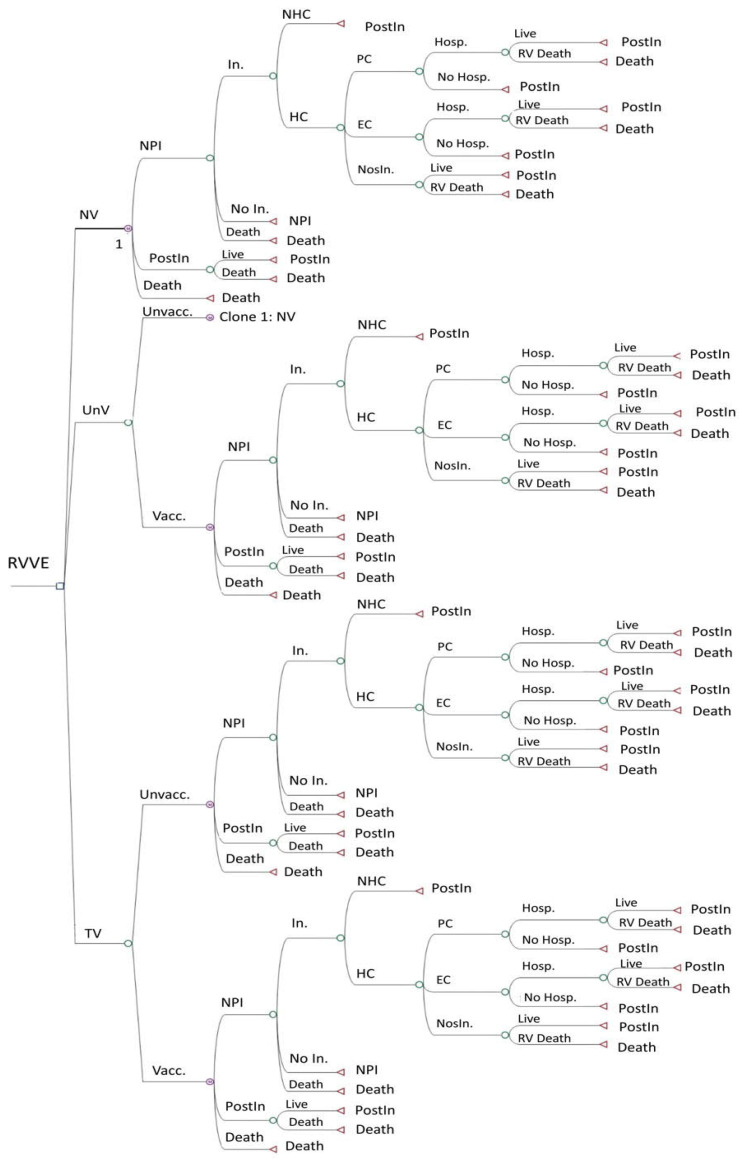
Decision tree of the Markov model for 3 vaccination strategies: no vaccination, universal vaccination, and targeted vaccination. EC: emergency care. HC: healthcare. Hosp.: hospitalization. In.: infection. NHC: no healthcare. NosIn.: nosocomial infection. NPI: no previous infection. NV: no vaccination. PC: primary care. PostIn: post-infection. RV: rotavirus. RVVE: rotavirus vaccination evaluation. TV: targeted vaccination. Unvacc.: unvaccinated. UnV: universal vaccination. Vacc.: vaccinated.

**Figure 2 viruses-16-01194-f002:**
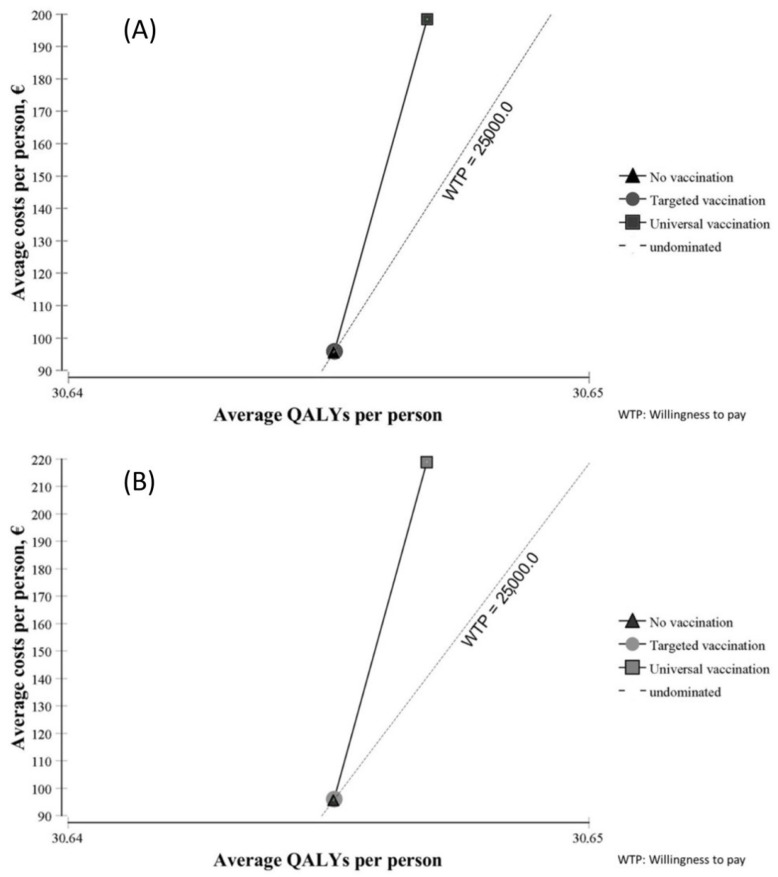
Cost–utility planes of Rotarix^®^ (**A**) and RotaTeq^®^ (**B**).

**Figure 3 viruses-16-01194-f003:**
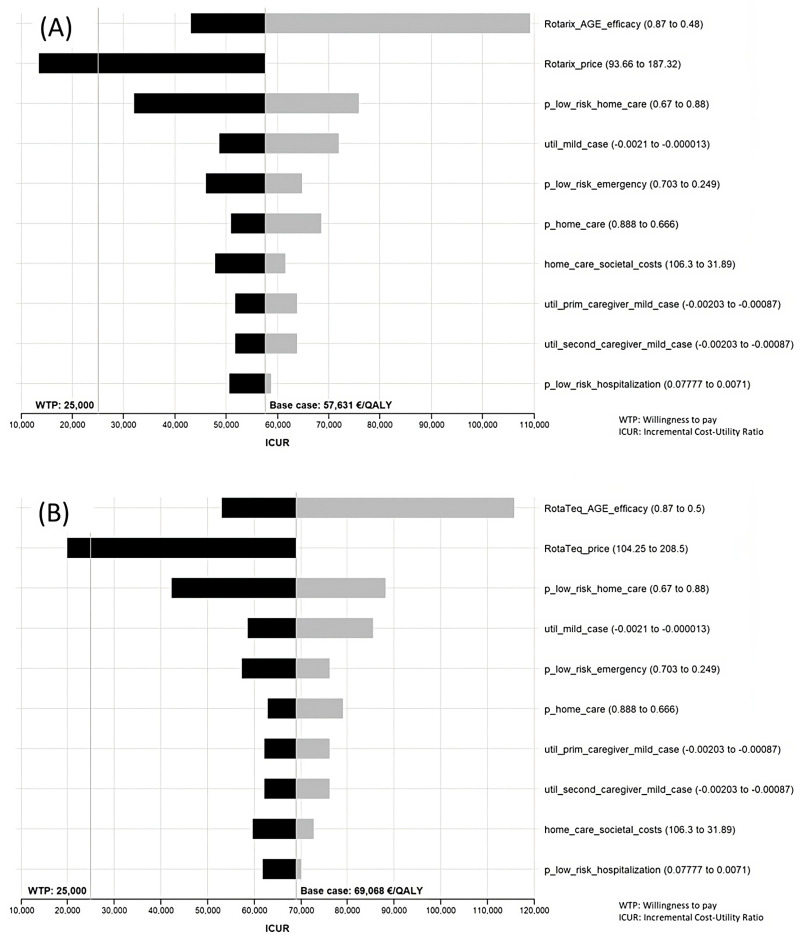
Tornado diagram. Sensitivity analysis of Rotarix^®^ (**A**) and RotaTeq^®^ (**B**) under societal perspective comparing universal vaccination with targeted vaccination.

**Table 2 viruses-16-01194-t002:** Utility losses attributable to each of the rotavirus cases that were included in the model.

	Rotavirus Case	QALY Lost	Source
Children	Mild case	0.0011	Aidelsburger, 2014 [23]
	Moderate case	0.0029	Marlow 2015 [22]
	Severe case	0.0034	
Primary caregiver	Mild case	0.0014	Hansen-Edwards 2017 [24]
	Moderate case	0.0014	Marlow 2015 [22]
	Severe case	0.004	
Secondary caregiver	Mild case	0.0014	Hansen-Edwards 2017 [24]
	Moderate case	0.0015	Marlow 2015 [22]
	Severe case	0.0028	

**Table 3 viruses-16-01194-t003:** Infections and related events produced and avoided with the use of Rotarix^®^.

	No Vaccination	Targeted Vaccination	Reduction with Targeted vs. No Vaccination	Universal Vaccination	Reduction with Universal vs. Targeted Vaccination
Deaths	0	0	0.0%	0	0.0%
Nosocomial infections	1030	1029	0.1%	366	64.4%
Hospitalizations	3841	3772	1.8%	1341	63.3%
Emergencies	22,839	22,332	2.2%	8233	61.7%
Primary care	29,814	29,621	0.6%	10,933	62.7%
Home care	215,625	214,712	0.4%	81,482	61.8%
Total infections	269,308	267,697	0.6%	101,014	61.9%

**Table 4 viruses-16-01194-t004:** Infections and related events produced and avoided with the use of RotaTeq^®^.

	No Vaccination	Targeted Vaccination	Reduction with Targeted vs. No Vaccination	Universal Vaccination	Reduction with Universal vs. Targeted Vaccination
Deaths	0	0	0.0%	0	0.0%
Nosocomial infections	1092	1026	6.0%	314	65.2%
Hospitalizations	4053	3849	5.0%	1209	65.1%
Emergencies	22,866	22,166	3.1%	7143	65.7%
Primary care	30,019	29,776	0.8%	9553	67.4%
Home care	215,075	214,352	0.3%	85,691	59.8%
Total infections	269,052	267,320	0.6%	102,701	61.2%

**Table 5 viruses-16-01194-t005:** Results of cost–utility analysis of both vaccines from a societal perspective.

	Average Cost per Person (EUR)	Incremental Cost (EUR)	Average Utility per Person (QALYs)	Incremental Utility (QALYs)	ICUR *
No vaccination	95.4571		30.6450905		
Targeted vaccination vs. no vaccination					
Targeted vaccination with Rotarix^®^	95.8896	0.4325	30.6451088	0.0000183	23,638
Targeted vaccination with RotaTeq^®^	96.0461	0.5891	30.6451089	0.0000184	32,008
Universal vaccination vs. no vaccination					
Universal with Rotarix^®^	198.3927	102.9357	30.6468874	0.0017969	57,285
Universal with RotaTeq^®^	218.8328	123.3758	30.6468866	0.0017962	68,687
Universal vaccination vs. targeted vaccination					
Universal with Rotarix^®^	198.3927	102.5032	30.6468874	0.0017786	57,631
Universal with RotaTeq^®^	218.8328	122.7867	30.6468866	0.0017778	69,068

* ICUR: Incremental Cost–Utility Ratio.

**Table 6 viruses-16-01194-t006:** Results of cost–utility analysis of both vaccines from an SNHS perspective.

	Average Cost per Person (EUR)	Incremental Cost (EUR)	Average Utility per Person (QALYs)	Incremental Utility (QALYs)	ICUR *
No vaccination	35.3236		30.6450905		
Targeted vaccination vs. no vaccination					
Targeted vaccination with Rotarix^®^	36.2592	0.9356	30.6451088	0.0000183	51,134
Targeted vaccination with RotaTeq^®^	36.4359	1.1123	30.6451089	0.0000184	60,436
Universal vaccination vs. no vaccination					
Universal with Rotarix^®^	176.2449	140.9214	30.6468874	0.0017969	78,424
Universal with RotaTeq^®^	197.6674	162.3438	30.6468867	0.0017962	90,382
Universal vaccination vs. targeted vaccination					
Universal with Rotarix^®^	176.2449	139.9857	30.6468874	0.0017786	78,705
Universal with RotaTeq^®^	197.6674	161.2315	30.6468867	0.0017778	90,692

* ICUR: Incremental Cost–Utility Ratio.

**Table 7 viruses-16-01194-t007:** Budget impact of extending from targeted to universal vaccination in Spain.

	Targeted Vaccination		Universal Vaccination with Current Prices		Universal Vaccination with Reduced Prices	
	Rotarix^®^	RotaTeq^®^	Rotarix^®^	RotaTeq^®^	Rotarix^®^	RotaTeq^®^
Population (newborns in Spain in 2021)	337,380	337,380	337,380	337,380	337,380	337,380
Vaccinated population	2945	2945	319,499	319,499	319,499	319,499
Prices of a complete vaccination (in EUR)	187.32	208.50	187.32	208.50	111.85–118.20	108.70–115.50
Administration costs (in EUR)	6.00	12.00	6.00	12.00	6.00	12.00
Gross annual budget for the SNHS (in EUR) *	569,391	649,445	61,765,520	70,449,499	37,652,941–39,681,758	38,563,512–40,736,105
Net annual budget for the SNHS (in EUR) **			61,196,129	69,800,054	37,083,550–39,681,758	37,914,068–40,086,660

* Annual cost of the vaccination program in 2020 (cost of the vaccination per person multiplied by the vaccinated population); ** budget increase that occurs when adding universal vaccination.

## Data Availability

All data generated or analysed during this study are included in this published article and its Appendix A. Further details are available from the corresponding author on reasonable request.

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
