# Peer review of "Updating and Refining of Economic Evaluation of Rotavirus Vaccination in Spain: A Cost–Utility and Budget Impact Analysis"

_viruses, 2024, doi:10.3390/v16081194_

Round 1

Reviewer 1 Report

Comments and Suggestions for Authors

The authors performed an update of a previous economic model on rotavirus vaccination in Spain published in 2014 in which they added now a comparison with a target vaccination policy currently recommended in the country. They also made a comparison between the two vaccines available on the Spanish market.

They used a conventional model construct of a Markov cohort model developed when the vaccine had to come to market with no long-term effect available at that time (2003).

The authors calculated here the cost-effectiveness and the budget impact trying to find out when universal mass vaccination against rotavirus in the country could be cost-effective under the threshold in Spain of 25000€ per QALY gained. The budget impact is limited to the extra budget needed for the vaccination moving from target to universal coverage.

The results show that target vaccination with one of the two vaccines available seen from a societal perspective can lead to a strategy that is cost-effective. If one looks closer to the effect data, there is not much of a big difference in the effect between the two vaccines. The ICER difference between the vaccines could therefore be compensated by a price adjustment.

I don’t have many comments to give because it is an update of an old model except that what the authors claimed as a budget impact analysis has a limited perspective on vaccine budget only. Not on all the other costs avoided or reduced due to the vaccination were assessed.

I have however some doubts about the targeted strategy approved (being cost-effective) whether that strategy adequately positions the vaccine role it may have. This should maybe be highlighted in the discussion section: if the market volume is very limited (<1% of the normal group (see Tabel 7)) as indicated here in Spain and in the Netherlands, it could be that producers are not interested in delivering the vaccine as that small amount was not their objective of their production capacity. Producers may therefore dramatically increase their price for getting out of their costs of goods…. Moreover, real life experience in the Netherlands indicates today that this approach of targeted group causes big logistical problems as they must deliver the small numbers of vaccines in much diverse places in time leading to poor adequate coverage. Given the vaccine to such a limited group, does not help to better control the infection problem the virus causes in general because of its contagion capacity. It is pushing a public health intervention into a treatment role without any effect of what vaccines normally can do to limit the spread.

This brings me to the next point. As the authors mentioned in the introduction, the vaccine is available since 2006 on the market.

The authors report about a review (Supplement 3) they did of the literature on cost-effectiveness data of the rotavirus-vaccines, but these were models published around the time that the vaccines were launched with high speculations about what the vaccines were expecting to do over time based on RTC data with a follow-up of 2y.

However, today there is experience of more than 15y with this vaccine reported in some countries in Europe that should be very useful to communicate back to decision makers who like to know the benefit short to long term of this vaccine in real life situations, now that those vaccines have been so long on the market. Maybe also to highlight in the discussion section in order to be complete in the assessment?

UK (2013) and Finland (2010) are quite successful with the vaccination strategy implemented with a current situation as if the infection is under best control. Other countries may have difficulties achieving the same results as measured over mid- to long-term (Belgium, 2006). There are different reasons for that lower long-term impact.

It is unfortunate that those publications are not covered here, whereas they give the useful information on how to become successful with this vaccination program which is not always obvious to achieve, as one needs to fulfill specific conditions of implementation so that the optimal vaccine effect is maintained over time for being only then very cost-effective!

It is fine to update an old model with new input-data and making new comparison, but it doesn’t tell anything about what the real benefit of this vaccine is about now, achieved by when and how, based on known real life experience. Previous studies from Spain reported vaccine coverage rates across the country that was very versatile. Those data have a big effect on the vaccine impact and cost-effectiveness results obtained (BMC ID, 2021, 21: 1138), also to be mentioned in this publication.

Author Response

We are very grateful for the comments and suggestions received, please find attached the document with our responses to the comments.

Reviewer 2 Report

Comments and Suggestions for Authors

Author Response

(The authors gave the same response as above.)

Reviewer 3 Report

Comments and Suggestions for Authors

Rotavirus is the leading cause of life-threatening gastroenteritis in infants and young children globally. Two live-attenuated rotavirus vaccines, Rotarix® and RotaTeq®, have been widely used and proven highly effective in reducing rotavirus caused diarrheal disease. In Spain, rotavirus vaccine is only funded for preterm babies born between 25 and 32 weeks of gestation but is not included in universal vaccination. In this manuscript, the authors assessed the cost-effectiveness of targeted and universal vaccination against rotavirus. The authors found only vaccination with Rotarix® in targeted infants from societal perspective is efficient, though it has minimal impact on the overall rotavirus disease burden. To make universal vaccination of Rotarix® and RotaTeq® efficient, price drop is required. 

This manuscript is well-written and provides much useful information, but still requires some minor modifications.

Minor comments:

1.        Font size in figure 1 is too small and barely readable. It would be helpful if the authors could increase the font size.

2.        Please check citation formatting. In some cases, there is a space between the quote and in-text citation number (e.g., line 38), which is redundant and would be deleted. 

3.        Line 239, two different fonts are used.

Author Response

(The authors gave the same response as above.)
